# Genotype by Environment Interaction Analysis for Grain Yield and Yield Components of Summer Maize Hybrids across the Huanghuaihai Region in China

Haiwang Yue [1] , Hugh G. Gauch [2], Jianwei Wei [1], Junliang Xie [1], Shuping Chen [1], Haicheng Peng [1], Junzhou Bu [1,*] and Xuwen Jiang [3,*]

[1] Dryland Farming Institute, Hebei Academy of Agriculture and Forestry Sciences, Hebei Provincial Key Laboratory of Crops Drought Resistance Research, Hengshui 053000, China; yanjiu1982@163.com (H.Y.); hengshuiwei@163.com (J.W.); hbhzsxjl@126.com (J.X.); hscsp@163.com (S.C.); yumiphc@163.com (H.P.)

[2] Soil and Crop Sciences, Cornell University, Ithaca, NY 14853, USA; hgg1@cornell.edu

[3] Maize Research Institute, College of Agronomy, Qingdao Agricultural University, Qingdao 266109, China

*  Correspondence: bujunzhou@126.com (J.B.); mjxw888@qau.edu.cn (X.J.)

**Abstract:** Increasing the maize production capacity to ensure food security is still the primary goal of global maize planting. The purpose of this study was to evaluate genotypes with high yield and stability in summer maize hybrids grown in the Huanghuaihai region of China using additive main effects and multiplicative interaction (AMMI) analysis and best linear unbiased prediction (BLUP) technique. A total of 18 summer maize hybrids with one check hybrid were used for this study using a randomized complete block design (RCBD) with three replicates at 74 locations during two consecutive years (2018–2019). A three-way analysis of variance (ANOVA) and an AMMI analysis showed that genotype (G), environment (E), year (Y) and their interactions were highly significant ($p < 0.001$) except G × E × Y for all evaluated traits viz., grain yield (GY), ear length (EL), hundred seed weight (HSW) and E × Y for hundred seed weight. The first seven interaction principal components (IPCs) were highly significant and explained 81.74% of the genotype by environment interaction (GEI). By comparing different models, the best linear unbiased prediction (BLUP) was considered the best model for data analysis in this study. The combination of AMMI model and BLUP technology to use the WAASB (weighted average of absolute scores from the singular value decomposition of the matrix of BLUP for GEI effects generated by linear mixed model) index was considered promising for similar research in the future. Genotypes H321 and Y23 had high yield and good stability, and could be used as new potential genetic resources for improving and stabilizing grain yield in maize breeding practices in the Huanghuaihai region of China. Genotypes H9, H168, Q218, Y303 and L5 had narrow adaptability and only apply to specific areas. The check genotype Z958 had good adaptability in most environments due to its good stability, but it also needs the potential to increase grain yield. Significant positive correlations were also found between the tested agronomic traits.

**Keywords:** *Zea mays* L.; AMMI model; BLUP; GEI; WAASB

## 1. Introduction

Maize (*Zea mays* L.) (2*n* = 20) is one of the most important crops on the earth, and the large-scale cultivation of maize plays a vital role in the continuous growth of the world's population [1–3]. Although a large number of crops have been planted and harvested all over the world, only four crops accounted for half of the global primary crop production in 2018, of which maize accounted for 13% of the total production with 1.1 billion tons, larger than rice (9%, 0.8 billion tons) and wheat (8%, 0.7 billion tons), while China produced about 25 percent of the world output of maize [4].

Field crops such as maize are often affected by environmental conditions. Therefore, some genotype responses are different depending on the genotype by environment interaction (GEI) effects formed by the joint actions of genotypes and environments [5–7]. The phenotype of different genotypes may be constant in various environments, while others show significant differences in different environments [8]. The key to improving agricultural production is to increase agricultural efficiency in the use of resources (increasing productivity per hectare and per dollar), which includes a better understanding of GEI and how it is used [6,9]. If there is no GEI, a single genotype of maize yields the highest worldwide, and the variety trials only need to be conducted in one location to provide universal results. It is necessary for plant breeders to examine the relationship between yield traits in order to determine appropriate selection criteria for breeding programs. The multi-environment trials (METs) are conducted annually on all major crops around the world, which are expensive but necessary to lead to the release and recommendation of new genotypes. METs are essential because genotype evaluation is complicated by differential responses of genotypes in different environments due to the presence of GEI [10–12].

In order to achieve the purpose of understanding and interpreting GEI, researchers have developed different statistical methods and have used these methods to study the grain yield and stability of maize genotypes [13–16]. Among the many analysis methods, the additive main effects and multiplicative interaction (AMMI) model is one of the most widely used methods in the analysis of METs [17]. The AMMI model uses analysis of variance (ANOVA) for additive or main effects followed by principal component analysis (PCA) for multiplicative or interactive effects. The graphical tools of this approach have special features in the simultaneous evaluation of trait performance and stability, as well as in the delineation of mega-environments and the selection of narrow adaptations. In addition, the AMMI model also has an extensive stability evaluation system [18,19]. The AMMI model has been widely used to analyze METs for two purposes, namely understanding the complex GEI and increasing accuracy [20]. Regarding the AMMI model, Gauch [21] reported a new program called AMMISOFT, which facilitates AMMI analysis to help accelerate crop improvement. This procedure has been widely used by researchers from various countries [22–24]. Apart from the advantages, the AMMI model also has shortcomings. For example, a sensitivity to the presence of individual outliers [25], and a lack of successful cases of linear mixed-effects model (LMM) analysis [26]. Olivoto et al. [26] merged the features of the AMMI model and the best linear unbiased prediction (BLUP) technique. One novel statistical parameter, WAASB (weighted average of absolute scores from the singular value decomposition of the matrix of best linear unbiased predictions for the genotype × environment interaction effects generated by linear mixed effect model), was brought for the selection of genotypes based on mean performance and stability. The combination of the AMMI graphical tool and the predictive accuracy of BLUP has been widely used in GEI studies [27–29].

The objectives of this study were to evaluate 19 representative summer maize hybrids in seven provinces, at 37 locations, and planted for 2 consecutive years (combination of year and location generated 74 environments), in order to study how yield components of summer maize hybrids were affected by the GEI and to identify genotypes with high yielding and stable performance, and the relationship between various yield components has also been studied.

## 2. Materials and Methods

### 2.1. Plant Materials, Locations and Experimental Design

In this study, 18 advanced maize genotypes and one check hybrid were studied for two years (2018 and 2019) at 37 locations across seven provinces based on a randomized complete block design (RCBD) with three replications. A characteristics and distribution map of the participating genotypes and individual locations are shown in Tables S1 and S2 and Figure S1. The plot at each location was composed of five rows with 0.6 m spacing between rows, and the area of each plot had 20.1 $m^2$ in size. Plant density was set at

75,000 plants per hectare for all evaluated hybrids in each location. The experimental data used in this study was a mixed yield data from the two years, 2018 and 2019. During the study period, the field managements for each location site were suited to local management measures without any obvious nutrient or water limitation.

### 2.2. Measurements

The agronomic traits measured in this study were as follows:

Grain yield (t/ha): When physiologically mature, an area of 12.1 m$^2$ was harvested manually (three rows in the center of each plot, 6.7 m long), and the grain weight was measured. The grain moisture content was measured with a portable moisture meter (PM8188, Kett Electric Laboratory, Tokyo, Japan). Grain yield was adjusting the moisture to 14% and converting the unit to tons per hectare.

Ear length (cm): At physiological maturity of the evaluated hybrids, 10 maize ears were manually harvested in the first row or the last row of each plot, and the ear length was measured from the bottom to the highest point, and the average number was obtained.

Hundred seed weight (g): A total of 100 maize seeds were randomly selected and weighed.

### 2.3. Statistical Analysis

#### 2.3.1. Linear Mixed Model

The yield components data were analyzed using the linear mixed model, represented by the following equation.

$$Y_{ger} = \mu + \alpha_g + \beta_e + (\alpha\beta)_{ge} + w_{er} + \varepsilon_{ger} \tag{1}$$

where $Y_{ger}$ is the observations of the yield components of genotype $g$ in the environment $e$ and block $r$, $\mu$ is the mean effect, $\alpha_g$ is the fixed effect of genotype $g$, $\beta_e$ is the random effect of environment $e$, $(\alpha\beta)_{ge}$ is the random effect interaction of genotype $g$ in environment $e$, $w_{er}$ is the random effect of block $r$ in environment $e$, and $\varepsilon_{ger}$ is the experimental error effect associated with the $g$th genotype, the $r$th block and the $i$th environment, which is assumed to be a normal independent distribution, with a mean of 0 and a variance of $\sigma^2$ [30]. The linear mixed model analysis was performed using the metan 1.14.0 packages [31].

#### 2.3.2. AMMI Model Analysis

The AMMI model integrates standard ANOVA and principal component analysis (PCA) to determine the interaction principal component (IPC) to calculate stability parameters. The AMMI model can be summarized by the following equation.

$$Y_{ger} = \mu + \alpha_g + \beta_e + \sum_n \lambda_n \gamma_{gn} \delta_{en} + \rho_{ge} + \varepsilon_{ger} \tag{2}$$

where $Y_{ger}$ represents the yield of genotype $g$ in environment e for replicate $r$; $\mu$ represents the grand mean, $\alpha_g$ and $\beta_e$ represent the genotypes and environments deviation from $\mu$, respectively; $\lambda_n$ represent the nth singular value of interaction principal component (IPC); $\gamma_{gn}$ and $\delta_{en}$ are the eigenvector values of genotype $g$ and environment $e$ of component $n$, respectively; $\rho_{ge}$ represents the residual of AMMI model; $\varepsilon_{ger}$ represents the error [32]. The GEI sum of squares (SS) related to "Noise" (GEI$_N$) is calculated by multiplying the mean square of the error and the degrees of freedom related to GEI, and then the GEI SS related to "Signal" (GEI$_S$) is calculated as the difference between the total GEI SS and GEI$_N$. A three-way ANOVA and AMMI analysis were conducted with the package *qdata* by R software version 4.0.1 (R Core Team, 2020, R Foundation for Statistical Computing, Vienna, Austria) and AMMISOFT version 1.0 (Soil and Crop Sciences, Cornell University, Ithaca, NY, USA), respectively. For statistical significance, we used the most reliable $F_R$ test, which can better optimize the prediction accuracy [33].



### 2.3.3. BLUP Technique

In the BLUP method, the effects of genotype and genotype by environment interaction (GEI) are considered to be random. Contrary to the AMMI model, a linear mixed model is used, and the formula is as follows:

$$Y = X\beta + Zu + \varepsilon \tag{3}$$

where $\beta$ is the data vector of the fixed unknown effect (the average value of the block in each environment), $u$ is the GEI + genotype effect, $X$ and $Z$ represent the matrix involving $\beta$, $u$ and $Y$, and $\varepsilon$ is the random errors' vector. In order to better predict the AMMI family model and the BLUP model, root mean square prediction difference (RMSPD) estimates were used to compare [34]. The variance components of agronomic traits were estimated by restricted maximum likelihood (REML) using the metan 1.14.0 package [31].

### 2.3.4. Combining of AMMI Analysis and BLUP Techniques

We used the method introduced by Olivoto et al. [26] in this study, combining the AMMI analysis and BLUP techniques. The stability index of each genotype in METs called WAASB (the weighted average of absolute scores from the singular value decomposition of the matrix of best linear unbiased predictions for the GEI effects generated by a linear mixed-effect model) index was calculated by the following formula:

$$WAASB_i = \frac{\sum_{k=1}^{p} |IPCA_{ik} \times EP_k|}{\sum_{k=1}^{p} EP_k} \tag{4}$$

where $WAASB_i$ is the weighted average of absolute scores of the $i$th genotype; $IPCA_{ik}$ is the score of the $i$th genotype in the $k$th interaction principal component axis (IPCA). In the usage of the traditional AMMI model, singular value decomposition (SVD) is used to decompose the matrix with additive model residuals into $k$ IPCAs, and scores are obtained by the SVD of the GEI effects obtained in the linear mixed effects model, and $EP_k$ is the amount of the variance explained by the $k$th IPCA. Genotypes with a lower WAASB value are considered to be more stable genotypes, on the contrary, a genotype with a higher WAASB value is generally considered to be unstable.

## 3. Results

### 3.1. The Prediction Accuracy of BLUP and AMMI Model

Root mean square prediction difference (RMSPD) is used to predict the accuracy of BLUP and AMMI models of the yield components, and the display results are based on the average of 200 predictions of RMSPD for each test model (Tables S3–S5). The model with the smallest RMSPD value is defined as the most accurate prediction, and vice versa. For grain yield, BLUP was the most accurate prediction model among all evaluated models. The highest genotypic mean based on BLUP prediction was L808 (11.2 t/ha), followed by L206 (11.1 t/ha), H9 (11.0 t/ha), Q218 (11.0 t/ha) and Y303 (10.9 t/ha) (Table S6). In addition to the above genotypes, the mean values of H110, H168, H321, Y23 and L5 were greater than the grand mean, and the mean values of the remaining nine genotypes were below the grand mean (Figure 1a). BLUP was found to be the most accurate model for predicting in terms of ear length (Table S4) and hundred seed weight (Table S5). Genotypes H9, H110, W702, D9, X20, Y23, L206 and N101 were the top eight hybrids in BLUP prediction. The mean performance for ear length of these eight genotypes were above the grand mean, and the predicted values of the remaining eleven genotypes were below the grand mean (Table S7 and Figure 1b). Similarly, BLUP was considered to be the most accurate model for the hundred seed weight prediction, and the predicted values of BLUP were presented in Table S8 and Figure 1c.

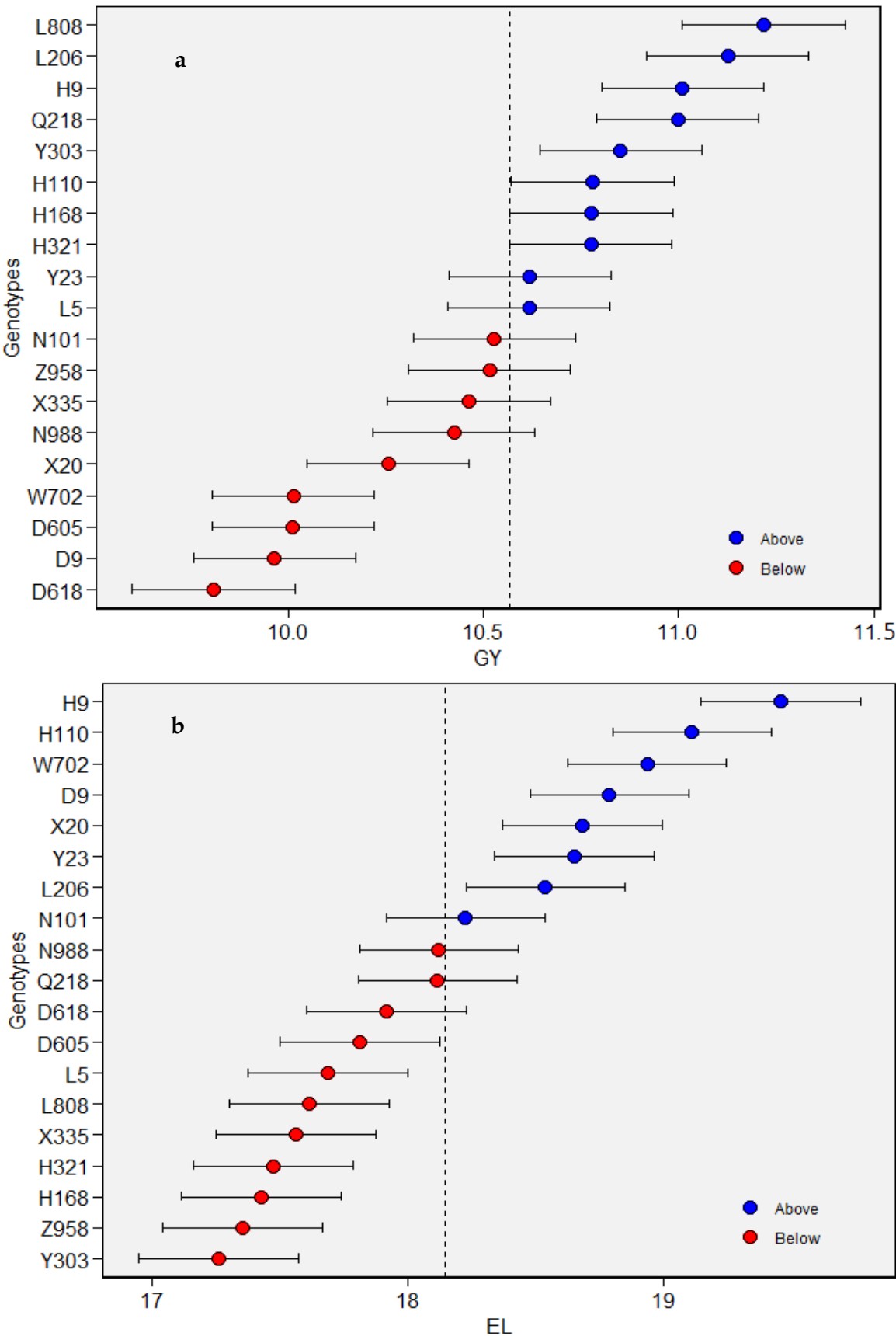

**Figure 1.** *Cont.*

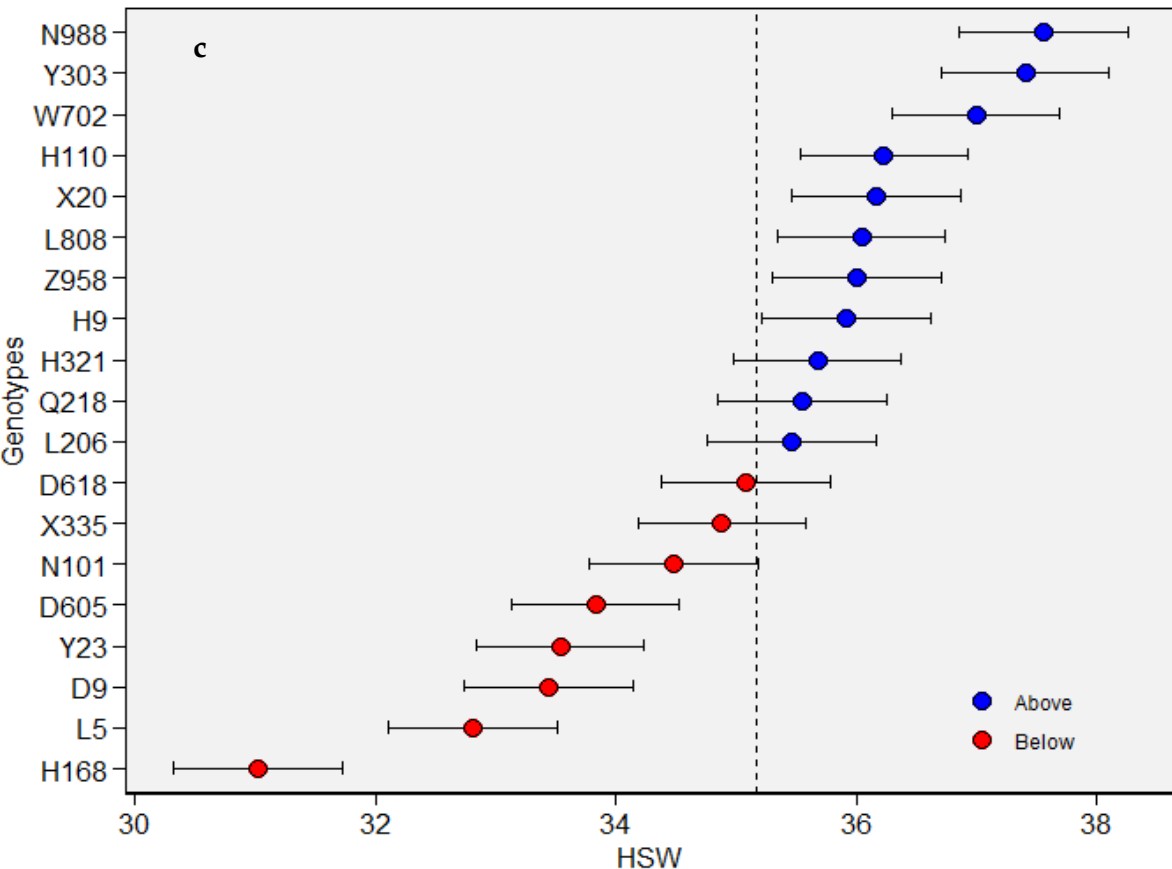

**Figure 1.** The predicted agronomic performance of 19 maize genotypes based on BLUP (best linear unbiased prediction). The blue and red circles represent higher and lower than the BLUP means, respectively. The horizontal error bars indicate the 95% confidence interval when considering the two-tailed *t*-test. (**a**) represent grain yield (GY); (**b**) represent ear length (EL); (**c**) represent hundred seed weight (HSW).

### 3.2. Variance Components of Yield Components

The likelihood ratio test results showed that the effects of genotype, environment and genotype by environment interactions in yield components were highly significant ($p < 0.001$) (Table 1). The proportions of the variance components in the phenotypic variance of the evaluated yield components were the same; that is, the variance of GEI effect ($\sigma_{ge}^2$) had the highest proportion, followed by the genotypic variance ($\sigma_g^2$), and the residual variance ($\sigma_\varepsilon^2$) had the lowest proportion. For grain yield, ear length and hundred seed weight, the observed estimates of broad-sense heritability ($H^2$) showed a lower level, which were 0.209, 0.230 and 0.261, respectively. In contrast, the genotypic accuracy of selection (As) in the above three traits were 0.976, 0.979 and 0.983, respectively. In addition, the high ratios of genotypic and residual coefficient of variation (CV ratio) for grain yield, ear length and hundred seed weight of 2.92, 2.72 and 1.63, respectively, made the genotype–environment correlation ($r_{ge}$) highly correlated (0.969, 0.960 and 0.867, respectively). The high proportion of the GEI effect in the yield components indicated that the GEI effect plays an important role in the expression of maize traits.

**Table 1.** Estimated variance components for three agronomic traits evaluated for 18 maize genotypes over 74 environments.

| | Likelihood Ratio Test | | | | | | | | |
|---|---|---|---|---|---|---|---|---|---|
| **Statistics** | **Grain Yield (t/ha)** | | | **Ear Length (cm)** | | | **Hundred Seed Weight (g)** | | |
| | **G** | **E** | **GE** | **G** | **E** | **GE** | **G** | **E** | **GE** |
| $\chi^2$ | 262 | 129 | 7808 | 300 | 136 | 7116 | 375 | 19.8 | 4025 |
| *p* value | $4.09 \times 10^{-19}$ | $5.31 \times 10^{-30}$ | 0 | $4.12 \times 10^{-27}$ | $1.71 \times 10^{-31}$ | 0 | $1.68 \times 10^{-18}$ | $8.64 \times 10^{-6}$ | 0 |
| REML | Estimated variance components | | | | | | | | |
| $\sigma_g^2$ | 0.1764 (20.9%) | | | 0.4486 (23.0%) | | | 2.839 (26.1%) | | |
| $\sigma_{ge}^2$ | 0.6484 (76.7%) | | | 1.442 (73.9%) | | | 6.977 (64.1%) | | |
| $\sigma_\varepsilon^2$ | 0.0207 (2.4%) | | | 0.0605 (3.1%) | | | 1.068 (9.8%) | | |
| $\sigma_p^2$ | 0.846 | | | 1.951 | | | 10.880 | | |
| $H^2$ | 0.209 | | | 0.230 | | | 0.261 | | |
| $R_{gei}^2$ | 0.767 | | | 0.739 | | | 0.641 | | |
| $h_{mg}^2$ | 0.952 | | | 0.958 | | | 0.966 | | |
| As | 0.976 | | | 0.979 | | | 0.983 | | |
| $r_{ge}$ | 0.969 | | | 0.960 | | | 0.867 | | |
| $CV_g\%$ | 3.98 | | | 3.692 | | | 4.792 | | |
| $CV_r\%$ | 1.36 | | | 1.355 | | | 2.940 | | |
| *CV* ratio | 2.92 | | | 2.724 | | | 1.630 | | |
| SD | 1.95 | | | 1.72 | | | 5.70 | | |
| SE | 0.03 | | | 0.03 | | | 0.09 | | |

Note: G, genotype; E, environment; GE, genotype by environment interaction; REML, restricted maximum likelihood; $\sigma_g^2$, genotypic variance; $\sigma_{ge}^2$, genotype by environment interaction variance; $\sigma_\varepsilon^2$, residual variance; $\sigma_p^2$, phenotypic variance; $H^2$, broad-sense heritability; $R_{gei}^2$, coefficient of determination of the interaction effects; $h_{mg}^2$, heritability of the genotypic mean; As, accuracy of selection; $r_{ge}$, genotype–environment correlation; $CV_g\%$, genotypic coefficient of variation; $CV_r\%$, residual coefficient of variation; *CV* ratio, ratio between genotypic and residual coefficient of variation; SD, standard deviation; SE, standard error.

### 3.3. Three-Way Analysis of Variance

The three-way analysis of variance (ANOVA) detected that the differences between the evaluated genotypes (G) were highly significant ($p < 0.001$), which proved that the hybrids used in this study had great differences in grain yield (Table 2). High significance ($p < 0.001$) was also observed between the environments (E) and years (Y). The three-way ANOVA showed that genotype effect, year effect, environment effect and their interactions had a highly significant impact on grain yield except for GEY, for which GEY had no significant effect. The highest proportion of variance was mainly affected by environment effect (61.74%), followed by genotype by environment interaction effect (14.40%), genotype effect (4.63%) and environment by year interaction effect (3.37%). The three-way analysis of variance also showed that except for the insignificant effects of GEY in ear length and EY and GEY in hundred seed weight, the other sources of variation reached significance at $p < 0.001$.

**Table 2.** The three-way analysis of variance for 19 maize genotypes for yield components during 2018 and 2019.

| Traits | Source of Variance | DF | SS | MS | F Value | Pr (>F) | Percent of Total SS (%) |
|---|---|---|---|---|---|---|---|
| Grain yield | Genotype (G) | 18 | 740.50 | 41.14 | 51.684 *** | <0.001 | 4.63 |
| | Environment (E) | 36 | 9872 | 274.20 | 344.50 *** | <0.001 | 61.74 |
| | Year (Y) | 1 | 22 | 22.45 | 28.21 *** | <0.001 | 0.14 |
| | G × E | 648 | 2303 | 3.56 | 3.56 *** | <0.001 | 14.40 |
| | G × Y | 18 | 130.9 | 7.27 | 9.14 *** | <0.001 | 0.82 |
| | E × Y | 36 | 538.3 | 14.95 | 18.79 *** | <0.001 | 3.37 |
| | G × E × Y | 648 | 148.1 | 0.228 | 0.29 ns | 1.00 | 0.93 |
| | Residuals | 2810 | 102.2 | 0.8 | | | |
| | Total | 4217 | 15,994.13 | | | | |
| Ear length | Genotype (G) | 18 | 1873.63 | 103.98 | 357.76 *** | <0.001 | 14.96 |
| | Environment (E) | 36 | 4037.21 | 112.14 | 385.97 *** | <0.001 | 32.23 |
| | Year (Y) | 1 | 19 | 19.07 | 66.16 *** | <0.001 | 0.15 |
| | G × | 648 | 5264.26 | 8.13 | 27.97 *** | <0.001 | 42.02 |
| | G × Y | 18 | 378.87 | 21.06 | 72.47 *** | <0.001 | 3.02 |
| | E × Y | 36 | 21.51 | 0.60 | 2.05 *** | <0.001 | 0.17 |
| | G × E × Y | 648 | 115.06 | 0.18 | 1.03 ns | 0.295 | 0.92 |
| | Residuals | 2810 | 483.11 | 0.17 | | | |
| | Total | 4217 | 12,527.07 | | | | |
| Hundred seed weight | Genotype (G) | 18 | 11,740.46 | 652.25 | 50.7 *** | <0.001 | 8.57 |
| | Environment (E) | 36 | 45,929.19 | 1275.81 | 99.18 *** | <0.001 | 33.52 |
| | Year (Y) | 1 | 5728.02 | 5728.02 | 445.29 *** | <0.001 | 4.18 |
| | G × E | 648 | 24,969.04 | 38.53 | 2.43 *** | <0.001 | 18.22 |
| | G × Y | 18 | 3861.11 | 214.51 | 16.68 *** | <0.001 | 2.82 |
| | E × Y | 36 | 45.18 | 1.25 | 0.1 ns | 1 | 0.03 |
| | G × E × Y | 648 | 84.13 | 0.13 | 0.01 ns | 1 | 0.06 |
| | Residuals | 2810 | 36,146.67 | 12.86 | | | |
| | Total | 4217 | 137,015.75 | | | | |

*** Significant at the 0.001 probability levels; ns, not significant at $p = 0.05$; DF, Degrees of freedom; SS, Sum of squares; MS, Mean square. The same as below.

### 3.4. AMMI Analysis of Variance

In the AMMI analysis, each year and location were combined into one environment, and it could be found that the environment (E) effect and the genotype by environment interaction (GEI) effect were 14.09 times and 3.49 times than the genotype effect, respectively, and the GEI effect was further partitioned into seven interaction principal components (IPCs), and all IPCs were highly significant ($p < 0.001$) (Table 3). The IPC1 explained 28.72% of the sum of squares of the GEI effect, while IPC2, IPC3, IPC4, IPC5, IPC6 and IPC7 explained 20.27, 9.73, 6.76, 6.25, 5.90 and 4.11%, respectively. Since AMMISOFT was limited to seven IPCs, the $F_R$ test showed that seven AMMI model families were determined. The estimated sums of squares for GEI$_S$ and GEI$_N$ accounted for 98.95% and 1.05% of the SS of GEI effect, respectively. The sum of squares for GEI$_S$ was 3.45 times that for G main effects; hence, narrow adaptations are important for this dataset. It can also be clearly seen from Table 3 that IPC1, IPC2 and IPC3 represent the AMMI model families AMMI1, AMMI2 and AMMI3, respectively, which have filled a total of 58.72% of GEI variation and 59.35% of GEI$_S$ variation.

**Table 3.** AMMI analysis for grain yield in evaluated maize genotypes during 2018 and 2019.

| Source of Variance | df | SS | MS | Proportion of Variation | | | |
| | | | | % of GE Signal and Noise | % of Variability Explained | % of GEI SS | % of GEIs Variation |
|---|---|---|---|---|---|---|---|
| Treatment | 1405 | 13,755.78 | 9.79 *** | | 86.01 | | |
| Genotype | 18 | 740.50 | 41.14 *** | | | | |
| Environment (E) | 73 | 10,432.84 | 142.92 *** | | | | |
| GE interaction (GEI) | 1314 | 2582.42 | 1.97 *** | | | | |
| IPC1 | 90 | 741.6 | 8.24 *** | | | 28.72 | 29.02 |
| IPC2 | 88 | 523.49 | 5.95 *** | | | 20.27 | 20.49 |
| IPC3 | 86 | 251.37 | 2.92 *** | | | 9.73 | 9.84 |
| IPC4 | 84 | 174.54 | 2.08 *** | | | 6.76 | 6.83 |
| IPC5 | 82 | 161.41 | 1.97 *** | | | 6.25 | 6.32 |
| IPC6 | 80 | 152.44 | 1.91 *** | | | 5.90 | 5.97 |
| IPC7 | 78 | 106.02 | 1.36 *** | | | 4.11 | 4.15 |
| Residual | 726 | 471.55 | 0.65 *** | | | 18.26 | 18.45 |
| Error | 2812 | 2238.34 | 0.80 | | 13.99 | | |
| Blocks/environment | 148 | 2183.21 | 14.75 *** | | | | |
| Pure Error | 2664 | 55.13 | 0.02 | | | | |
| $GEI_N$ | | 27.19 | | 1.05 | | | |
| $GEI_S$ | | 2555.23 | | 98.95 | | | |
| Total | 4217 | 15,994.13 | 3.79 | 100 | 100 | 100 | 100 |

*** Significant at the 0.001 probability levels; $GEI_S$, estimated sums of squares for G × E signal; $GEI_N$, estimated sums of squares for G × E noise; IPC, interaction principal component.

### 3.5. WAASB Scores of Evaluated Genotypes

In this study, the stability of the yield components of the tested genotypes were evaluated based on the WAASB scores. Genotype Y23 (0.25) was considered to be the most stable followed by Z958 (0.3), H321 (0.341), L5 (0.383) and Y303 (0.409), while genotype D9 (1.08) was found to be the most unstable, followed by W702 (0.811), D618 (0.720) and L808 (0.686) for grain yield (Table 4). Genotypes with the lowest WAASB scores were H168 (0.365), followed by N988 (0.439), H321 (0.465), H110 (0.556) and Y303 (0.585). Thus, H168, N988, H321, H110 and Y303 were found to be stable genotypes in terms of ear length (Table 5). For hundred seed weight, a lower value of WAASB measure was observed for X20 (0.562), X335 (0.737), Q218 (0.764), D618 (0.774) and N988 (0.814); it was also found that H168 (1.47), D9 (1.29), W702 (1.15) and N101 (1.13) were genotypes with poor stability (Table 6).

**Table 4.** The mean grain yield and WAASB scores of evaluated genotypes.

| Genotype Code | Grain Yield (t/ha) | PC1 | PC2 | PC3 | PC4 | PC5 | PC6 | PC7 | WAASB | rWAASB |
|---|---|---|---|---|---|---|---|---|---|---|
| D605 | 9.98 (17) | −0.459 | −1.58 | 0.135 | −0.181 | 0.33 | −0.451 | 0.147 | 0.594 | 12 |
| D618 | 9.77 (19) | 1.440 | −0.109 | −0.49 | −1.88 | −0.692 | −0.030 | −0.355 | 0.72 | 17 |
| D9 | 9.93 (18) | 2.6 | 0.658 | 0.196 | 0.932 | 0.326 | −0.741 | 0.011 | 1.08 | 19 |
| H110 | 10.8 (6) | 0.696 | −0.572 | 0.266 | 0.976 | −1.89 | 0.296 | −0.111 | 0.595 | 13 |
| H168 | 10.8 (7) | −0.993 | 0.384 | 0.124 | 0.351 | 0.542 | 0.044 | −0.462 | 0.538 | 9 |
| H321 | 10.8 (8) | −0.312 | −0.558 | −0.093 | 0.402 | 0.018 | 0.623 | 0.15 | 0.341 | 3 |
| H9 | 11 (3) | −0.923 | 0.48 | 0.472 | −0.162 | −0.433 | 0.318 | 0.54 | 0.575 | 11 |
| L206 | 11.2 (2) | −0.790 | 0.738 | −0.053 | −0.083 | −0.774 | −1.300 | 1.06 | 0.615 | 14 |
| L5 | 10.6 (10) | 0.0793 | −0.524 | −0.297 | 0.381 | 0.998 | −0.943 | 0.112 | 0.383 | 4 |
| L808 | 11.3 (1) | −0.591 | 0.976 | 1.700 | −0.864 | 0.223 | −0.129 | −0.183 | 0.686 | 16 |
| N101 | 10.5 (11) | −0.258 | −1.34 | 0.439 | 0.206 | 0.216 | 0.483 | −0.2 | 0.546 | 10 |
| N988 | 10.4 (14) | 0.557 | −1.66 | 0.215 | −0.314 | 0.398 | −0.064 | 0.417 | 0.666 | 15 |
| Q218 | 11 (4) | −0.052 | 1.031 | 0.582 | 0.456 | 0.142 | −0.353 | −0.923 | 0.451 | 6 |
| W702 | 9.98 (16) | 1.220 | 1.072 | −0.264 | −0.199 | 0.786 | 1.38 | 0.717 | 0.811 | 18 |
| X20 | 10.2 (15) | −0.667 | 0.493 | −1.27 | 0.063 | 0.053 | 0.0875 | 0.92 | 0.529 | 8 |

**Table 4.** *Cont.*

| Genotype Code | Grain Yield (t/ha) | PC1 | PC2 | PC3 | PC4 | PC5 | PC6 | PC7 | WAASB | rWAASB |
|---|---|---|---|---|---|---|---|---|---|---|
| X335 | 10.5 (3) | −0.507 | 0.284 | −1.700 | −0.257 | −0.026 | −0.372 | −1.12 | 0.52 | 7 |
| Y23 | 10.6 (9) | −0.0229 | −0.283 | 0.433 | −0.479 | −0.047 | 0.0571 | −0.318 | 0.25 | 1 |
| Y303 | 10.9 (5) | −0.616 | 0.128 | −0.366 | 0.489 | −0.161 | 0.78 | −0.491 | 0.409 | 5 |
| Z958 | 10.5 (12) | −0.404 | 0.391 | −0.024 | 0.162 | −0.014 | 0.317 | 0.083 | 0.3 | 2 |

PC, interaction principal component; WAASB, weighted average of absolute scores from the singular value decomposition of the matrix of best linear unbiased predictions for the genotype × environment interaction effects generated by linear mixed effect model; rWAASB, genotype ranking based on WAASB scores. The same below.

**Table 5.** The mean ear length and WAASB scores of evaluated genotypes.

| Genotype Code | Ear Length (cm) | PC1 | PC2 | PC3 | PC4 | PC5 | PC6 | PC7 | WAASB | rWAASB |
|---|---|---|---|---|---|---|---|---|---|---|
| D605 | 17.8 (12) | −0.683 | 0.636 | −0.569 | 0.241 | −0.815 | 0.591 | 0.477 | 0.6 | 6 |
| D618 | 17.9 (11) | −0.054 | 1.24 | 0.596 | 0.057 | −0.999 | −0.962 | 1.05 | 0.607 | 9 |
| D9 | 18.8 (4) | −0.119 | −1.6 | 1.73 | −0.21 | 0.586 | −0.415 | −0.577 | 0.709 | 15 |
| H110 | 19.2 (2) | −0.792 | −0.912 | −0.256 | 0.35 | 0.019 | −0.342 | 0.0479 | 0.556 | 4 |
| H168 | 17.4 (17) | −0.417 | −0.027 | −0.568 | −0.045 | −0.253 | 0.668 | −0.372 | 0.365 | 1 |
| H321 | 17.4 (16) | 0.254 | 0.194 | −0.581 | 0.114 | −1.26 | 0.439 | −0.232 | 0.465 | 3 |
| H9 | 19.5 (1) | −1.27 | 0.055 | −0.478 | 0.585 | −0.9 | −1.15 | −0.739 | 0.751 | 17 |
| L206 | 18.6 (7) | 1.22 | −2.25 | −1.03 | −0.673 | −0.007 | −0.285 | −0.040 | 0.789 | 19 |
| L5 | 17.7 (13) | 0.852 | 0.914 | −0.277 | 0.51 | 1.48 | 1.22 | 0.178 | 0.703 | 14 |
| L808 | 17.6 (14) | 0.595 | 0.123 | −1.34 | −1.9 | −0.415 | −0.165 | −0.147 | 0.602 | 8 |
| N101 | 18.2 (8) | −1.81 | −0.227 | 0.424 | 0.33 | −0.21 | 0.227 | 1.17 | 0.677 | 10 |
| N988 | 18.1 (9) | 0.717 | −0.255 | −0.020 | 0.734 | 0.007 | 0.583 | 0.621 | 0.439 | 2 |
| Q218 | 18.1 (10) | 0.731 | 1.66 | 0.001 | 0.253 | 0.752 | −1.17 | −1.77 | 0.757 | 18 |
| W702 | 19.0 (3) | 0.808 | −0.056 | 2.43 | −0.156 | −0.653 | 0.267 | −0.263 | 0.679 | 11 |
| X20 | 18.7 (5) | −1.21 | 0.185 | −0.053 | −1 | 1.39 | −1.3 | 0.552 | 0.734 | 16 |
| X335 | 17.5 (15) | −0.756 | 0.532 | 0.22 | −1.16 | 1.39 | 1.05 | 0.342 | 0.682 | 12 |
| Y23 | 18.7 (6) | −0.028 | −0.62 | −0.706 | 2.27 | 0.847 | −0.006 | −0.256 | 0.6 | 7 |
| Y303 | 17.2 (19) | −0.401 | 0.094 | 0.355 | −0.48 | −0.778 | 1.37 | −1.27 | 0.585 | 5 |
| Z958 | 17.3 (18) | 2.36 | 0.312 | 0.12 | 0.177 | −0.175 | −0.617 | 1.22 | 0.699 | 13 |

**Table 6.** The mean hundred seed weight and WAASB scores of evaluated genotypes.

| Genotype Code | Hundred Seed Weight (g) | PC1 | PC2 | PC3 | PC4 | PC5 | PC6 | PC7 | WAASB | rWAASB |
|---|---|---|---|---|---|---|---|---|---|---|
| D605 | 33.8 (15) | −0.386 | −1.63 | −0.697 | 2.1 | 0.645 | −0.163 | 0.873 | 0.974 | 11 |
| D618 | 35.1 (12) | 0.776 | 0.139 | 1.9 | −0.789 | 0.513 | −1.19 | −0.429 | 0.774 | 4 |
| D9 | 33.4 (17) | −3.5 | −1.21 | 0.951 | −1.13 | −0.325 | 0.563 | 0.314 | 1.29 | 18 |
| H110 | 36.3 (4) | −1.01 | 2.81 | −0.539 | 0.846 | −0.329 | −1.89 | 0.269 | 1.12 | 15 |
| H168 | 30.9 (19) | 2.24 | −2.34 | 1.24 | −0.346 | −2.42 | 1.14 | 1.01 | 1.47 | 19 |
| H321 | 35.7 (9) | 1.48 | 0.784 | 1.82 | −0.202 | 0.186 | −1.22 | −0.837 | 0.934 | 9 |
| H9 | 35.9 (8) | 1.16 | 0.398 | −1.24 | −1.08 | −1.18 | −0.553 | 1.78 | 0.984 | 12 |
| L206 | 35.5 (11) | 0.941 | −0.529 | −1.51 | 0.41 | −0.701 | −1.55 | 0.212 | 0.898 | 8 |
| L5 | 32.7 (18) | −1.61 | −1.5 | −0.094 | 0.945 | −0.447 | −1.47 | −0.333 | 0.942 | 10 |
| L808 | 36.1 (6) | 1.04 | −1.45 | −1.65 | 0.427 | 2.31 | −0.207 | 1.04 | 1.11 | 13 |
| N101 | 34.5 (14) | −0.143 | 1.42 | 0.55 | 3.09 | −2.08 | 1.7 | −0.729 | 1.13 | 16 |
| N988 | 37.6 (1) | 0.111 | 0.168 | 0.667 | 1.42 | 2.93 | 1.42 | −0.112 | 0.814 | 5 |
| Q218 | 35.6 (10) | 0.614 | −0.203 | −1.13 | −1.92 | 0.191 | 0.378 | −0.668 | 0.764 | 3 |
| W702 | 37.1 (3) | −0.473 | 3.12 | 0.843 | −0.791 | 0.419 | 0.873 | 2.52 | 1.15 | 17 |
| X20 | 36.2 (5) | 0.356 | 0.619 | −0.045 | −0.302 | −0.116 | −1.09 | −1.93 | 0.562 | 1 |
| X335 | 34.9 (13) | −0.26 | 0.256 | −2.5 | 0.0338 | −0.59 | 1.22 | −1.08 | 0.737 | 2 |
| Y23 | 33.5 (16) | −3.05 | −1.14 | 0.706 | −0.96 | −0.151 | −0.059 | −0.037 | 1.11 | 14 |
| Y303 | 37.5 (2) | 0.207 | 0.959 | −1.16 | −1.73 | 0.317 | 1.71 | −1.36 | 0.826 | 7 |
| Z958 | 36.0 (7) | 1.5 | −0.666 | 1.89 | −0.0212 | 0.833 | 0.384 | −0.51 | 0.819 | 6 |

### 3.6. Integrate AMMI Model and BLUP Technology to Understand the GEI

According to the abscissa representing the yield components and the ordinate representing the WAASB value, the biplot can be divided into four different quadrants (Figure 2). Genotypes D9, W702, D618, D605, N101, X335 and X20 and environments GX18, MZ18, MZ19 and SS19 are located in the first quadrant; these genotypes and environments performed lower than the mean grain yield, and the contribution to the GEI is greater. The genotypes in the first quadrant are defined as genotypes with low yield and poor stability. The environments had a strong ability to distinguish genotypes. In the second quadrant, genotypes L808, L206, H110, H9, H168, Q218, Y303 and L5 and environments QZ18, QZ19, SX18, SX19, GX19, FP19, JH19, JZ19, XY19, JS19, HX19, FC19 and LY19 were present. The genotypes in this quadrant can be considered to have a higher grain yield, but similar to the first quadrant, it plays a larger role in the GEI. The environments contained in this quadrant deserve attention because, in addition to providing higher grain yield performance, they also have good discrimination capabilities to the genotypes. Environments MC18, MC19, XH18, XH19, SZ18, SZ19, HD18, HD19, WY18, WY19, GC18, GC19, BT18, BT19, LZ18, LZ19, SS18, MJ18, SP18, YL19, QS19, SP19 and MJ18, and only the genotype Z958 were placed in the third quadrant. The genotypes in this quadrant have a lower grain yield and a higher stability. Moreover, the environments that belong to this quadrant have the lowest WAASB value among all environments, which makes these environments have a poor ability to distinguish genotypes. The corresponding remaining genotypes and environments fall into the fourth quadrant. The genotypes in this quadrant have more than the mean grain yield performance and better stability (lower WAASB value). The environments in this quadrant have a higher grain yield but with a poor ability to distinguish genotypes (Figure 2a).

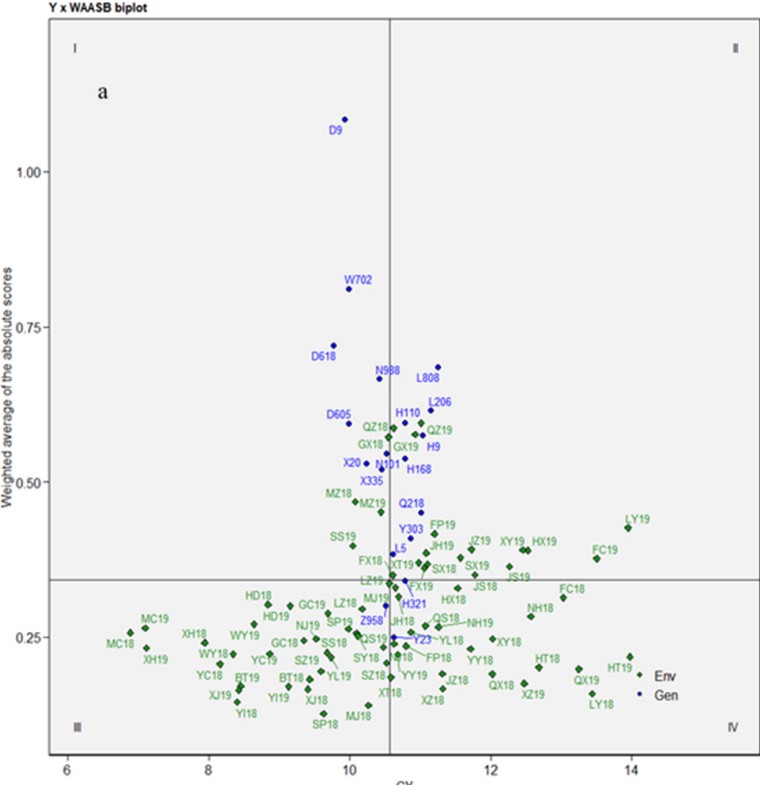

**Figure 2.** *Cont.*

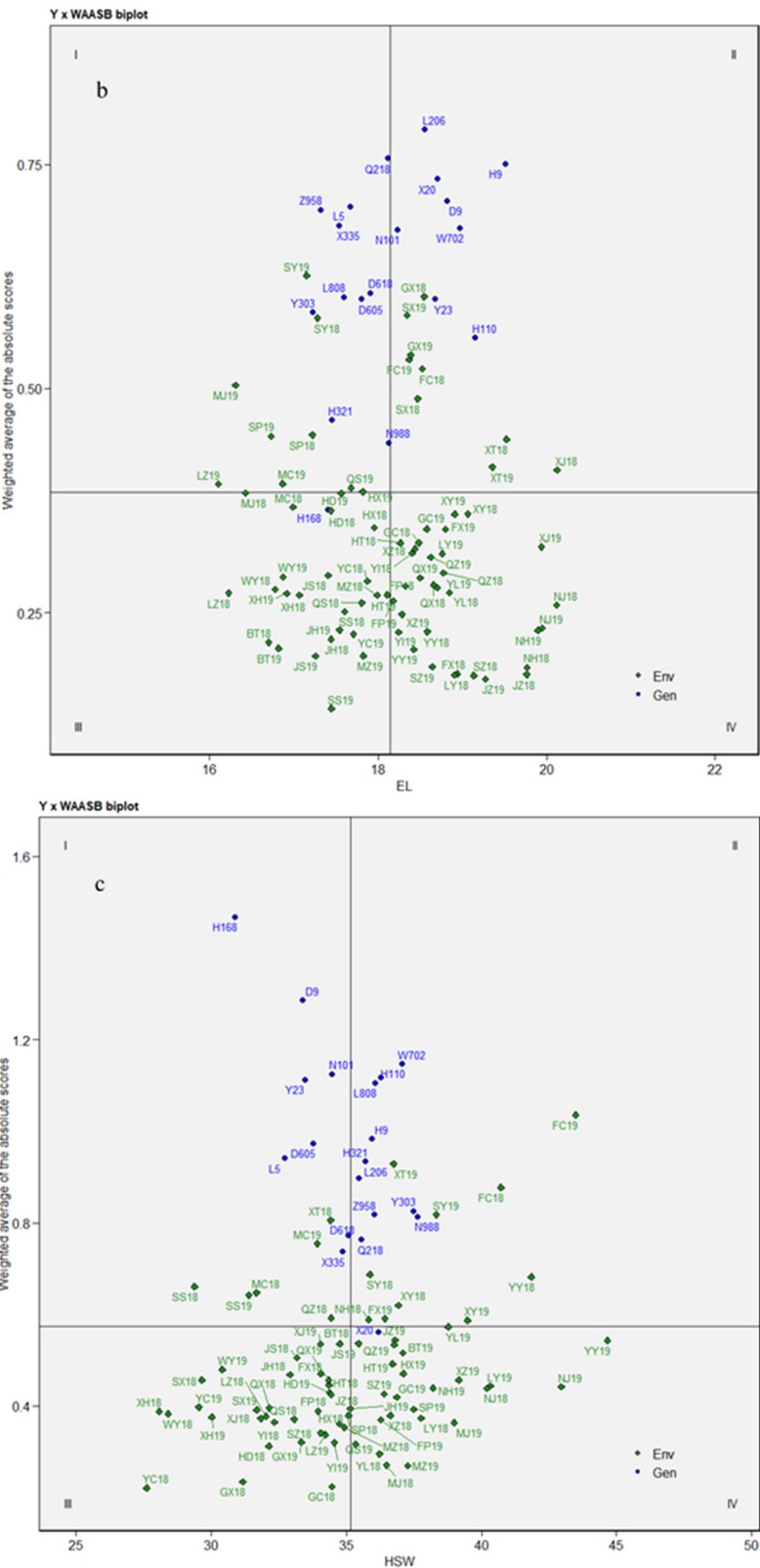

**Figure 2.** The biplot of agronomic traits combined with weighted average of absolute scores for the best linear unbiased predictions of the genotype vs. environment interaction (WAASB). GY, grain yield; EL, ear length; HSW, hundred seed weight. The blue and green icons represent the codes of environment and genotype, respectively. (**a**) represent grain yield (GY); (**b**) represent ear length (EL); (**c**) represent hundred seed weight (HSW).

The first quadrant includes highly unstable genotypes and highly discriminating environments. Genotypes Q218, Z958, L5, X335, Y303, L808, D618, D605, H321 and N988 being the unstable genotypes with lower than mean ear length in this quadrant, and environments SY18, SY19, SP18, SP19, MJ19, LZ19, MC19 and QS19 were found to effectively distinguish the tested genotypes. Unstable but highly productive genotypes and environments are included in the second quadrant. Genotypes L206, H9, X20, D9, N101, W702, Y23 and H110 were unstable genotypes, but had a mean greater than grand mean. Environments in this quadrant such as GX18, SX19, GX19, FC19, FC18, SX18, XT18, XT19 and XJ18 were discriminative with the environmental means were higher than the grand mean. Contrary to the second quadrant, the genotypes and environments in the third quadrant were low-productive, stable and wide-adapted owing to the lower scores of WAASB. Among the tested genotypes, only H168 and environments viz., SS19, MZ19, JS19, BT18 and BT19 fall into this quadrant. Finally, in the fourth quadrant, which contains extensive adaptations and productive genotypes with low WAASB scores. The environments contained in this quadrant can be considered productive but have low discrimination abilities (Figure 2b).

The first quadrant contains the most unstable genotypes W702, H110, L808, H9, H321, L206, Z958, Y303 and N988, and these genotypes were below the grand mean. Similar to the first quadrant, the genotypes in the second quadrant were also unstable, but their hundred seed weight values were good. Genotypes H168, D9, N101, Y23, D605, L5, D618 and X335 were present. No genotype falls into the third quadrant, and X20 was divided into the fourth quadrant (Figure 2c).

*3.7. Correlation and Cluster Analyses*

It was found that the various traits among the yield components have reached an extremely significant level ($p < 0.001$) based on Pearson's correlation (Figure S2). According to Figure S2, grain yield was positively associated with ear length ($r = 0.14$) and hundred seed weight (0.20). Significantly positive correlation ($r = 0.17$) was also found between ear length and hundred seed weight. The 19 tested genotypes in this study were divided into three clusters based on Euclidean distance for grain yield. The genotypes included in the first cluster were X20, X335, D9, D618 and W702. The second cluster contained the same number of genotypes as the first cluster, including D605, N988, L5, N101 and Y23. The third cluster included nine genotypes, L206, H9, L808, H321, Y303, H168, Z958, H110 and Q218 (Figure S3). Similarly, 74 environments used in this study were divided into four clusters according to Euclidean distance. The first cluster was made up of twenty environments, namely MC18, MC19, HX18, HX19, SP18, XY18, SS18, SS19, MZ18, MZ19, HD18, HD19, YL18, YL19, FP18, FP19, QZ18, QZ19, GX18 and GX19. The second cluster consisted of HT18, HT19, XT18, XT19, SZ18, SZ19, BT18, BT19, JZ18, WY18, WY19, YY18, YY19, XJ18, XJ19, QX18, QX19, XZ18, XZ19, YI18 and YI19. The environments NJ18, NJ19, JH19, SY19, LY18, FC19, LY19, SP19, MJ18, JZ19, MJ19, LZ18, LZ19, GC18, GC19, XY19, JS18, JS19, NH18 and NH19 were classified as the third cluster, and the rest of the environments were divided into the fourth cluster (Figure S4).

## 4. Discussion

The identification of yield components of maize hybrids, analysis of genotype by environment interaction (GEI) models and evaluation of yield stability are very important for the selection of highly productive and broadly adapted genotypes [35]. The GEI studies about maize genotypes are mainly focused on the AMMI model [36,37] and GGE biplot [38,39]. There are still few studies on evaluating the GEI interaction effects of maize yield components using WAASB as a stability parameter. The accurate prediction of the models is helpful for effective analysis and interpretation of MET research [18]. A BLUP-based mixed model has proven to be more accurate than the fixed effects AMMI model in many cases [30,40]. In addition, breeders can use the advantages of mixed models to



analyze genotypes in single- or multi-environment trials with variance components and genetic parameter estimation.

The three-way ANOVA and AMMI analysis showed that the evaluated maize hybrids had great variability, which indicated that these genotypes had differences in yield performance; therefore, further adaptability and stability analysis is required for commercial planting. In addition, the large changes in the interpretation of environmental (E) effects (65.23% of total SS) indicated that the environment was diverse and that the huge differences between environmental measures had caused most of the changes in grain yield. A small part of the total sum of squares caused by the treatment was attributed to the genotype (G) effects, and the magnitude of the genotype by environment interaction (GEI) sum of squares was higher than the G effect, indicating that there were considerable differences in genotype responses across different environments. Almost all similar studies have reported such results [39,41,42].

The importance of using narrow adaptability can be reflected in the high GEIs on $SS_G$, and the high $GEI_N$ on $SS_G$ indicates that primary IPCs should be used to improve accuracy because they selectively capture signals. AMMI is not just a single model, but constitutes a series of models, from AMMI0 to AMMIF. AMMI0 does not capture $GEI_N$ and GEIs, while AMMIF is a complete model. It has no residuals and captures all $GEI_N$ and $GEI_S$. Therefore, model selection occupies the most important position in AMMI analysis, and model diagnosis can provide clues for selecting the best model family for the existing dataset [21,42]. The AMMI analysis of this study shows that the first seven IPC are highly significant, cumulatively covering 81.74% of GEI variations and 82.62% of $GEI_S$ variations. Therefore, studying the distribution of the evaluated genotypes and environments on the basis of these seven components can provide helpful information for maize breeders [43].

If the variance explained in the first two interaction principal components (IPCs) is relatively low, then the interpretation of the traditional additive main effects and multiplicative interaction (AMMI) model will be biased, because most of the GEI explained by the remaining IPCs is not used. In order to show that AMMI1 information can be used reliably, we introduced the WAASB index, which is based on BLUP technology and includes all IPCs [26]. The first two IPCs can explain most of the genotype variation, but for some genotypes, more IPCs is necessary to explain the variation. For this reason, the WAASB value that considers all the significant IPCs can be regarded as a quantitative strategy of stability to interpret these variations. In this study, IPC1 only explained 29.02% of the GEI variation. In view of the reasons that some abiotic factors were not considered and used in this study, only seven IPCs were obtained. The WAASB index can be used to explain the variance explained in the axes other than IPC1. The WAASB stability index has been used to identify yield traits in different crops, such as wheat [44], soybean [45], lentils [46] and rice [47]. It was found that the grain yield of genotypes Y23 and H321 were highly stable in different environments using this model, and they also found that their grain yield was higher than the grand mean. In the GY × WAASB biplot, these two genotypes were in the fourth quadrant, meaning that they had good performances of grain yield and stability. We should also pay attention to the performance of some genotypes with high yield and not very poor stability, for example, genotypes H9, H168, Q218, Y303 and L5 can show their high yield potential in certain regions.

Studies on the evolution of yield components in China have shown that ear length and hundred seed weight are consistent with the significant increase in grain yield over time [48,49]. In the current study, there is a significant positive correlation between grain yield, ear length and hundred seed weight, which is consistent with previous reports [50–52]. The direct positive selection with ear length and hundred seed weight will simultaneously increase grain yield of maize genotypes. Parents of genotypes classified into the same group in cluster analysis can play a role in cross utilization, thereby generating greater genetic variability in segregating populations, and helping to breed inbred maize with better yield components.

## 5. Conclusions

In this study, 19 maize genotypes were evaluated in a two-year (2018–2019) field experiment, which was conducted at 37 locations in the Huanghuaihai region of China using the AMMI model and BLUP technique, and the three-way analysis of variance clearly demonstrated that the yield components of evaluated maize genotypes was highly affected by genotype (G), environment (E), year (Y) and interaction between these three effects, except G × E × Y and E × Y for hundred seed weight. The combination of the AMMI model and BLUP technique made it possible to describe GEI effects more accurately. In the present study, high-yield and stable genotypes, such as genotypes Y23 and H321, could be used as new potential genetic resources for improving and stabilizing grain yield in China. Genotypes H9, H168, Q218, Y303 and L5 only had narrow adaptability to special environments. The check hybrid Z958 had good stability and little potential for grain yield. It was also found that there was a highly significant positive correlation between the evaluated three agronomic traits.

**Supplementary Materials:** The following supporting information can be downloaded at: https://www.mdpi.com/article/10.3390/agriculture12050602/s1. The Supplementary Materials contains a total of eight tables and four figures. Table S1: Details of evaluated maize genotypes and their parentages used in this study. Table S2: Description of agro-climatic characteristics of environments used in the study during 2018–2019. Table S3: Predictive accuracy of AMMI model and BLUP by RMSPD estimates for grain yield. Table S4: Predictive accuracy of AMMI model and BLUP by RMSPD estimates for ear length. Table S5: Predictive accuracy of AMMI model and BLUP by RMSPD estimates for hundred seed weight. Table S6: Predictive value of BLUP genotype effect on grain yield. Table S7: Predictive value of BLUP genotype effect on ear length. Table S8: Predictive value of BLUP genotype effect on hundred seed weight. Figure S1: The main distribution map of the evaluated genotypes samples in China. Figure S2: The Pearson's correlation between the different agronomic traits used in this study. Figure S3: Cluster analysis of tested genotypes for grain yield based on a dissimilarity matrix. Figure S4: Cluster analysis using euclidean distance for grain yield among environments.

**Author Contributions:** H.Y.: Conceptualization, Data curation, Formal analysis, Funding acquisition, Investigation, Methodology, Software, Visualization, Writing—original draft, Writing—review and editing; H.G.G.: Conceptualization, Statistical analysis, Writing—review and editing; J.W.: Conceptualization, Writing—original draft, Writing—review and editing; J.X.: Conceptualization, Data curation, Formal analysis, Funding acquisition, Methodology, Supervision; S.C.: Conceptualization, Data curation, Formal analysis, Funding acquisition, Methodology, Supervision, Validation; H.P.: Conceptualization, Data curation, Funding acquisition, Methodology, Supervision; J.B. and X.J.: Conceptualization, Data curation, Formal analysis, Funding acquisition, Investigation, Methodology, Software, Visualization, Writing—original draft, Writing—review and editing. All authors have read and agreed to the published version of the manuscript.

**Funding:** This study was funded by the Key Research and Development Projects of Hebei Province (20326305D), Special Fund for National System (Maize) of Modern Industrial Technology (CARS-02), the Science and Technology Support Program of Hebei Province (16226323D-X), "Three-Three-Three Talent Project" Funded Project in Hebei Province (A202101056), HAAFS Science and Technology Innovation Special Project, the National Key Research and Development Program of China (2019YFE0120400) and Natural Science Foundation of Shandong Province, China (ZR2021MC107).

**Institutional Review Board Statement:** Not applicable.

**Informed Consent Statement:** Not applicable.

**Data Availability Statement:** The dataset analyzed during the current study is available from the corresponding author upon reasonable request.

**Conflicts of Interest:** The authors declare no conflict of interest.

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
