# Peer review of "Genotype by Environment Interaction Analysis for Grain Yield and Yield Components of Summer Maize Hybrids across the Huanghuaihai Region in China"

_agriculture, doi:10.3390/agriculture12050602_

Round 1

Reviewer 1 Report

The Ms of this version was significantly improved, and models used in this research were clearly tested. Although small grammar mistakes were found, for me the MS can be considered for publishment once minor revision were refined.

  1. in the PDF version I could not found the figure caption and related explaination for the figure one.
  2. in line 423, replace the two references with reference codes.
  3. line 437-438: "only seven IPCs......", other possible reason could be that some abiotic factors were not considered and used in this study.
  4. Figures S3 and S4 were not mentioned in the result part but only discussed in lines 454-457 (perhaps?).

Author Response

We feel great thanks for your professional review work on our manuscript. As you are concerned, there are several problems that need to be addressed. According to your nice suggestions, we have made some corrections to our previous draft, the detailed corrections are listed below.

Question 1. in the PDF version I could not found the figure caption and related explaination for the figure one.

Response:

In the original manuscript, the title and content of Figure 1 could not be presented due to the problem of picture setting. We have re-edited Figure 1 in the new manuscript version.

Question 2. in line 423, replace the two references with reference codes.

Response:

I totally agree with the reviewer’s suggestion. According to the reviewer's suggestion, we have normalized editing of these two references in the new manuscript version.

Question 3. line 437-438: "only seven IPCs......", other possible reason could be that some abiotic factors were not considered and used in this study.

Response:

I totally agree with the reviewer’s suggestion. Following the reviewer's suggestion, we have changed this sentence to "In view of the reasons that some abiotic factors were not considered and used in this study, only 7 IPCs were obtained."

Question 4. Figures S3 and S4 were not mentioned in the result part but only discussed in lines 454-457 (perhaps?).

Response:

In the previous version, we forgot the content of correlation and cluster analysis, and in this new version, we have made corresponding supplements.

3.7 Correlation and cluster analyses

It was found that the various traits among the yield components have reached an extremely significant level (P<0.001) based on Pearson’s correlation (Figure S2). Ac-cording to Figure S2, grain yield was positively associated with ear length (r=0.14***) and hundred seed weight (0.20***). Significantly positive correlation (r=0.17***) was also found between ear length and hundred seed weight. The 19 tested genotypes in this study were divided into 3 clusters based on euclidean distance for grain yield. The genotypes included in the first cluster were X20, X335, D9, D618 and W702. The second cluster contained the same number of genotypes as the first cluster, including D605, N988, L5, N101 and Y23. The third cluster included 9 genotypes, L206, H9, L808, H321, Y303, H168, Z958, H110 and Q218 (Figure S3). Similarly, 74 environ-ments used in this study were divided into 4 clusters according to euclidean distance. The first cluster was made up of twenty environments, namely MC18, MC19, HX18, HX19, SP18, XY18, SS18, SS19, MZ18, MZ19, HD18, HD19, YL18, YL19, FP18, FP19, QZ18, QZ19, GX18 and GX19. The second cluster consisting of HT18, HT19, XT18, XT19, SZ18, SZ19, BT18, BT19, JZ18, WY18, WY19, YY18, YY19, XJ18, XJ19, QX18, QX19, XZ18, XZ19, YI18 and YI19. The environments NJ18, NJ19, JH19, SY19, LY18, FC19, LY19, SP19, MJ18, JZ19, MJ19, LZ18, LZ19, GC18, GC19, XY19, JS18, JS19, NH18 and NH19 were classified as the third cluster, and the rest of the environments were divided into the fourth cluster (Figure S4).

Reviewer 2 Report

I am of the opinion that the manuscript has been significantly improved so that it can now be accepted for publication with minor technical corrections.

Author Response

Thank you again for your positive comments on our manuscript. According to the reviewer’s comments, we have made corresponding revisions to the manuscript.

This manuscript is a resubmission of an earlier submission. The following is a list of the peer review reports and author responses from that submission.

Round 1

Reviewer 1 Report

This article checked the differences of grain yield and environment stability among 19 maize hybrids in China using the genotype by environment interaction analysis. The authors found significant differences of the tested hybrids in genotype, enironment and year factors with ANOVA. The authors also performed an AMMI model to check the interactions between genotypes and environments, and found some hybrid maize genotypes having high grain yield and good stability that could be used as potentials for planting resource.

As this research had used statistic model and analysis on multiple parameters obtained from designed plots, basic statistic data were indispensable, and need to be clearly presented to illustrate the results and discussion. However, in the article, the experiment parameters were not clearly described in the method part, and some of the results were not easy to be identified referring to their description. The points listed below could be considered by the authors:

Abstract:
Line 23-24: in the research only the AMMI model was used for data analysis, how could the author emphasize that this model is suitable for their data without any comparison on other statistic models?

Introduction:
Description of this part seemed disordered and might need to be rewritten and rearranged, for example, lines 47-55 had explained the importance of GEI for maize yields but were redundant. The authors also mentioned the AMMI model and referred program, while this part was suggested to be addressed and moved to the methods part.

Materials and methods:
This part plays an important role in the statistical analysis but requires detailed explanation. For example the explanation of measurement of grain yield. What index was used to represent the grain yield of the hybrid samples in this research? When and how was the grain yield been measured? Did the authors use a mixed yield data from the two years 2018 and 2019? or these data were operated separately?

Line 92: is there any difference between the one checked hybrid and other 18 tested maize genotypes? Please note the checked hybrid sample in table 1. Also, a distribution map of the genotype samples is suggested, which seems to be more clear to readers than the table 1.

Table 2: what is the environment factor used in the designed plots and statistic analysis? For one location, there were two environment codes listed, is there any difference between the two?

Results:
Line 136: according to table 3, the genotype effect only explained 4.63% of the total yield differences that seems not "great" among these hybrids. Additionally, as there were significant differences among the tested genotypes based on table 3, the basic data and statistics for the grain yield and environment factors in each hybrid location are suggested to be supplied.

Table 4: according to the description in the method and the result parts, the GEIn and GEIs could be GEn and GEs, respectively?

Line 157: please explain the "FR test" as it was not even mentioned in the method part.

Figures 1, 2 and 3 are suggested to be improved by the authors if possible, as some of the results could not be well identified and might lead to misunderstandings to readers.

Lines 257-259: these results were not well reflected or shown in the figure 4.

Discussion:
The authors gave a global summary for their data and statistical analysis without any clear reference to their obtained results. So it seemed hard for readers to get the points discussed in this part. In addition, the authors are suggested to concentrate their main discussion on the differences among the tested genotypes and potenial utilization in the maize planting in China.

Author Response

Dear reviewer,

We really appreciate your careful review of this manuscript entitled “Genotype by environment interaction analysis for grain yield of summer maize hybrids across Huanghuaihai region in China” (Manuscript ID: agriculture-1623330). We have fully adopted the comments and made corresponding revisions to the revised manuscript.

I hope to be recognized by the reviewer, and I also look forward to inviting you to China to give constructive guidance to our work. If I can listen to your teachings in China, I believe it will have guiding significance for our future research work.

Thank you and best regards.

Yours sincerely,

Professor Haiwang Yue

Now I will giving a point-by-point response to the reviewers' comments.

Abstract:

Question 1. Line 23-24: in the research only the AMMI model was used for data analysis, how could the author emphasize that this model is suitable for their data without any comparison on other statistic models?

Response:

In the last version, we only used the one agronomic trait, grain yield, for analysis. As the reviewer 3 said, analyzing only one trait of grain yield would cause the loss of the analysis. In this new manuscript version, we have added ear length (EL) and hundred seed weight (HSW) combined with grain yield, a total of 3 agronomic traits for analysis.

I totally agree with the reviewer’s comment. In the previous version of the manuscript, we only used one model, the AMMI model. Indeed, as the reviewer 1 said, it is impossible to explain whether the model selection is suitable in the case of only 1 model. In the revised version, we have added a new model, the best linear unbiased prediction (BLUP) model, based on the root mean square prediction difference (RMSPD) value as the criterion to compare the accuracy of the AMMI and BLUP models. The model with the smallest RMSPD value is defined as the most accurate prediction, and vice versa. BLUP was the most accurate prediction model among all evaluated agronomic traits.

Introduction:

Question 2. Description of this part seemed disordered and might need to be rewritten and rearranged, for example, lines 47-55 had explained the importance of GEI for maize yields but were redundant. The authors also mentioned the AMMI model and referred program, while this part was suggested to be addressed and moved to the methods part.

Response:

I totally agree with the reviewer’s suggestion. According to the reviewer's suggestion, we have reorganized and supplemented the introduction part of the article. Also, we have revised the explanation part of the AMMI model program.

Materials and methods:

Question 3. This part plays an important role in the statistical analysis but requires detailed explanation. For example the explanation of measurement of grain yield. What index was used to represent the grain yield of the hybrid samples in this research? When and how was the grain yield been measured? Did the authors use a mixed yield data from the two years 2018 and 2019? or these data were operated separately?

Response:

I totally agree with the reviewer’s comment on materials and methods of manuscript. The materials and methods section introduced in the last version was a bit simplistic, and we have reorganized this section in the newly revised manuscript. Several issues raised by the reviewer have been well represented in the revised manuscript. In the revised manuscript, we have added a new measurements section in the Materials and methods section to describe the  methods for the 3 agronomic traits measured in this study.

Grain yield (t/ha): When physiologically mature, an area of 12.1 m2 were harvested manually (three rows in the center of each plot, 6.7 m long), and the grain weight was measured. The grain moisture content was measured with a portable moisture meter (PM8188, Kett Electric Laboratory, Japan). Grain yield was adjusting the moisture to 14% and converting the unit to tons per hectare.

Ear length (cm): At physiological maturity of evaluated hybrids, 10 maize ears were manually harvested in the first row or the last row of each plot, and the ear length was measured from the bottom to the highest point, and the average number was obtained.

Hundred seed weight (g): 100 maize seeds were randomly selected and weighed.

The experimental data used in this study was a mixed yield data from the two years 2018 and 2019.

Question 4. Line 92: is there any difference between the one checked hybrid and other 18 tested maize genotypes? Please note the checked hybrid sample in table 1. Also, a distribution map of the genotype samples is suggested, which seems to be more clear to readers than the table 1.

Response:

The check genotype Zhengdan 958 used in this study is a hybrid like the other 18 maize genotypes. This hybrid is the unified check hybrid for the current Muti-environment trials (METs) in the Huanghuaihai region of China. Zhengdan 958 had been on the market for many years and had made a huge contribution to China's food supply. Now the new hybrids selected by breeders used this hybrid as a check hybrid, the purpose is to improve the yield, stability, disease resistance and other indicators beyond it. We have flagged the check hybrid in Table S1 of the revised manuscript, and have also added a sample distribution map of the evaluated genotypes.

Question 5. Table 2: what is the environment factor used in the designed plots and statistic analysis? For one location, there were two environment codes listed, is there any difference between the two?

Response:

In this study, we selected 37 locations in 7 provinces in the Huanghuaihai region of China and planted for 2 consecutive years (combination of year and location generated 74 environments). In the three-way analysis of variance, genotype effect, environment effect, and year effect are used as three factors of variation, and the environmental factor is 37 pilots per year, so the degrees of freedom of environment are 36 (37-1=36). Whereas in AMMI ANOVA, the environmental factor is a combination of location and year, so the degree of freedom for the environmental factor is 73 (74-1).

The 18 and 19 following the same environment code represent the years 2018 and 2019, respectively. We put 18 and 19 after the location code based on how the locations changed from year to year.

Results:

Question 6. Line 136: according to table 3, the genotype effect only explained 4.63% of the total yield differences that seems not "great" among these hybrids. Additionally, as there were significant differences among the tested genotypes based on table 3, the basic data and statistics for the grain yield and environment factors in each hybrid location are suggested to be supplied.

Response:

I totally agree with the reviewer’s comment. Based on the reviewer's suggestion, we have added a new table titled “Select specific variables and compute statistics by levels of a factor variable for environment factor”. In this table, we have added some basic data and statistical analysis of 3 tested traits in 74 environments in 2018 and 2019. These indicators include CV: coefficient of variation; Max: maximum value; Min: minimum value; Mean: arithmetic mean; Sd.amo: the sample standard deviation; SE: the standard error of the mean; CI: 95 percent confidence interval of the mean, this table is presented separately as an Excel table. Here, I would like to ask, because this form is relatively long, can it appear in the supplementary tables and figures section in the form of a table?

Question 7. Table 4: according to the description in the method and the result parts, the GEIn and GEIs could be GEn and GEs, respectively?

Response:

The GEn and GEs that appeared in the previous version of the manuscript were GEIn and GEIs, respectively, and we unify them in the new version.

Question 8. Line 157: please explain the "FR test" as it was not even mentioned in the method part.

Response:

AMMI (Additive Main Effect and Multiplicative Interaction) stability model is a graph-based GEI modeling tool, but it cannot accommodate linear mixed effect models in the structure. BLUP (best linear unbiased prediction) can provide reliable response estimates, but it is not a graph-based tool for dealing with random GEI structures. In order to combine the characteristics of these two models, the quantitative stability index WAASB (the weighted average of absolute scores from the singular value decomposition of the BLUP matrix of the GEI effect generated by LMM) based on LMM (linear mixed model) was developed to study GEI through biplot standardization. WAASB is a mixed-model form of AMMI considering genotypes as random variables and taking into account all the IPCs (interaction principal component) for stability analysis. AMMI Stability Value (ASV), most commonly used stability index is based on squared deviations. On contrary, WAASB index is based on absolute deviations resulting in enhanced robustness and insensitivity to outliers (Olivoto et al., 2019). Based on the above principles, we added the BLUP model prediction and the application of the stability index WAASB to the revised manuscript, and classified and evaluated the tested genotypes in the form of GY×WAASB biplots. As for the issue of the method of statistical significance mentioned by the reviewer, we have added “For statistical significance, we used the most reliable FR test, which can better optimize the prediction accuracy to the method section in the revised manuscript. This test was implemented for the RCBD data as described in Piepho [33]. Reference used as follows:

Hadasch, S.; Forkman, J.; Piepho, H.P. Cross‐Validation in AMMI and GGE Models: A Comparison of Methods. Crop Sci. 2017, 57, 264-274.

Question 9. Figures 1, 2 and 3 are suggested to be improved by the authors if possible, as some of the results could not be well identified and might lead to misunderstandings to readers.

Response:

Figures 1, 2 and 3 in the old version have been removed in the new version due to our add of new agronomic traits and models in the revised manuscript.

Question 10. Lines 257-259: these results were not well reflected or shown in the figure 4.

Response:

In the same way as question 9, Figure 4 in the old version has also been deleted in the new version.

Discussion:
Question 11.
The authors gave a global summary for their data and statistical analysis without any clear reference to their obtained results. So it seemed hard for readers to get the points discussed in this part. In addition, the authors are suggested to concentrate their main discussion on the differences among the tested genotypes and potenial utilization in the maize planting in China.

Response:

I totally agree with the reviewer's opinion. We have re-edited and reorganized the discussion section in accordance with the reviewer's comments.

Reviewer 2 Report

Detailed comments and remarks are given in the manuscript file.

Author Response

Dear reviewer,

We greatly appreciate the careful review of Reviewer 2. We have fully adopted the comments of Reviewer 2 and made corresponding revisions to the revised manuscript.

The latest revision can be seen in the attachment

Best wishes.

Reviewer 3 Report

The manuscript entitled “Genotype by environment interaction analysis for grain yield of summer maize hybrids across Huanghuaihai region in China” focused on the screening of different genotypes of maize, based on the phenotypic data. For that reason, the authors have applied a combined analysis technique of ANOVA and the AMMI model. I appreciate the efforts of the authors. However, I do not find any specific objective and scientific significance in this manuscript.

If- the research focuses on the statistical analysis, then the aspect will be different, and then, authors should try to develop a new model or comparisons between the different models for a specific analysis or improve a specific model.

If- the research focuses on the screening of genotypes, then this could be a part of a complete manuscript (I mean, it is not a complete manuscript). For genotype screening, we generally applied at least two of the following techniques; phenotype analysis, molecular analysis, physiological analysis, and biochemical analysis. Where authors used only phenotypic data. For phenotypic screening, we always say that at least two years of data are essential. Two years of phenotypic data are considered sufficient when you have molecular analysis data. Phenotypes are the most complex, and two years of phenotypic data is not sufficient for phenotypic screening without molecular analysis.

The depth of study is very poor, for example,

If – we move the Tables 1, 2 and 6 in supplementary, the manuscript will not have any data.

[line 340-342] the three-factor analysis of variance clearly……. Three effects except G×E×Y. This is known knowledge and nothing new in the findings of the current experiment.
Yes, I can understand that this specific analysis might be helpful for the next (advanced) study of your genetic material. However, publishing a manuscript using only this data and data analysis is not a good idea.

Author Response

Dear reviewer,

we would like to express our sincere thanks to the Reviewer 3 for the evaluation of the manuscript. According to reviewer 3's suggestion on the manuscript, we have added models comparison in the revised manuscript, introduced BLUP and WAASB methods, and added two agronomic traits indicators, ear length and hundred seed weight. In particular, the statistical methods of the manuscript are thoroughly strengthened. I hope to be recognized by the reviewer, and I also look forward to inviting you to China to give constructive guidance to our work. If I can listen to your teachings in China, I believe it will have guiding significance for our future research work.

The latest revision can be seen in the attachment

Best wishes.
